# ICOSL Stimulation by ICOS-Fc Accelerates Cutaneous Wound Healing In Vivo

**DOI:** 10.3390/ijms23137363

**Published:** 2022-07-01

**Authors:** Ian Stoppa, Casimiro Luca Gigliotti, Nausicaa Clemente, Deepika Pantham, Chiara Dianzani, Chiara Monge, Chiara Puricelli, Roberta Rolla, Salvatore Sutti, Filippo Renò, Renzo Boldorini, Elena Boggio, Umberto Dianzani

**Affiliations:** 1Department of Health Sciences and Interdisciplinary Research Center of Autoimmune Diseases (IRCAD), Università del Piemonte Orientale, 28100 Novara, Italy; ian.stoppa@uniupo.it (I.S.); luca.gigliotti@med.uniupo.it (C.L.G.); nausicaa.clemente@med.uniupo.it (N.C.); deepika.pantham@uniupo.it (D.P.); 20032501@studenti.uniupo.it (C.P.); roberta.rolla@med.uniupo.it (R.R.); salvatore.sutti@med.uniupo.it (S.S.); filippo.reno@med.uniupo.it (F.R.); renzo.boldorini@med.uniupo.it (R.B.); umberto.dianzani@med.uniupo.it (U.D.); 2NOVAICOS srls, 28100 Novara, Italy; 3Department of Scienza e Tecnologia del Farmaco, University of Turin, 10124 Turin, Italy; chiara.dianzani@unito.it (C.D.); chiara.monge@unito.it (C.M.); 4Maggiore della Carità University Hospital, 28100 Novara, Italy

**Keywords:** ICOS:ICOSL system, wound healing, reparative macrophages

## Abstract

Background: ICOS and its ligand ICOSL are immune receptors whose interaction triggers bidirectional signals that modulate the immune response and tissue repair. Aim: The aim of this study was to assess the in vivo effects of ICOSL triggering by ICOS-Fc, a recombinant soluble form of ICOS, on skin wound healing. Methods: The effect of human ICOS-Fc on wound healing was assessed, in vitro, and, in vivo, by skin wound healing assay using ICOS^−/−^ and ICOSL^−/−^ knockout (KO) mice and NOD-SCID-IL2R null (NSG) mice. Results: We show that, in wild type mice, treatment with ICOS-Fc improves wound healing, promotes angiogenesis, preceded by upregulation of IL-6 and VEGF expression; increases the number of fibroblasts and T cells, whereas it reduces that of neutrophils; and increases the number of M2 vs. M1 macrophages. Fittingly, ICOS-Fc enhanced M2 macrophage migration, while it hampered that of M1 macrophages. ICOS^−/−^ and ICOSL^−/−^ KO, and NSG mice showed delayed wound healing, and treatment with ICOS-Fc improved wound closure in ICOS^−/−^ and NSG mice. Conclusion: These data show that the ICOS/ICOSL network cooperates in tissue repair, and that triggering of ICOSL by ICOS-Fc improves cutaneous wound healing by increasing angiogenesis and recruitment of reparative macrophages.

## 1. Introduction

Skin wound healing starts immediately after injury and evolves in three phases. The first one is an inflammatory phase during which platelets tend to aggregate, while inflammatory cells are recruited to the wound site. The second proliferative phase is characterized by the formation of granulation tissue and re-epithelialization due to the migration and proliferation of keratinocytes, fibroblasts, and ECs, and by ECM deposition. The last one is the so-called remodeling phase during which the regenerative process comes to an end and the wound becomes avascular and acellular, thereby allowing the reorganization of the connective tissue to promote scar formation [1,2].

ICOS (CD278) is a T cell co-stimulatory receptor, member of the CD28 family [3], mainly expressed on activated T-cells. ICOS binds ICOSL (CD275, also called B7h, GL50, B7H2), a member of the B7 family. ICOS triggering in T cells promotes not only the activation of effector T cells in peripheral tissues but also the development of regulatory T cells [4]. ICOSL is expressed on multiple cell types, including antigen presenting cells (APCs), activated ECs, epithelial cells, fibroblasts, and keratinocytes [5,6]. ICOSL triggering mediated by ICOS drives a “reverse signal” that inhibits the migration of endothelial, dendritic, and tumor cells, modulates cytokine secretion while promoting antigen cross-presentation in dendritic cells, and inhibits osteoclast differentiation and functions [7,8,9,10,11].

We have recently shown that ICOSL also binds osteopontin (OPN) at a different site from that used to bind ICOS [12], which suggests that the ICOSL/OPN axis may play a role in wound healing besides tumorigenesis. This hypothesis is also supported by the observation that OPN can act as both an ECM component and a soluble cytokine involved in inflammation and angiogenesis [13,14]. Indeed, ICOSL triggering by OPN induces tumor cell migration and promotes tumor angiogenesis, both of which are counteracted by ICOS-mediated activation of ICOSL [12].

The formal demonstration of a functional role of the ICOS/ICOSL pathway in wound healing comes from the observation that ICOS^−/−^, ICOSL^−/−^, and ICOS/ICOSL^−/−^ mice show delayed wound healing [15] likely due to decreased production of IL-6 [16]. In good agreement with a role of the ICOS/ICOSL dyad in normal tissue repair, we have recently shown that CCl_4_-induced liver damage, which is dependent on massive recruitment of blood-derived monocytes/macrophages, is dramatically worsened in both ICOS^−/−^ and ICOSL^−/−^ mice [17]. Interestingly, we were able to rescue this impairment by treating mice with ICOS-Fc, a recombinant soluble protein composed of the ICOS extracellular portion fused to the IgG1 Fc portion, which has been previously shown to trigger ICOSL, thereby inhibiting the development of experimental tumor metastases in vitro and tumor angiogenesis in vivo [9,11,17,18].

As the aforementioned findings support a functional role of ICOS/ICOSL in tissue repair, in the present study, we sought to determine the effect of ICOS-Fc in both in vitro and in vivo models of skin wound healing. Our in vivo results show that ICOS-Fc improves would healing likely by increasing angiogenesis and recruitment of reparative macrophages.

## 2. Results

### 2.1. ICOSL Activation by ICOS-Fc Increases Keratinocyte Migration In Vitro

To begin to explore the role of ICOSL in tissue repair, we assessed the effect of human ICOS-Fc on keratinocyte wound healing in vitro by scratch assay on HaCat human keratinocytes, which are known to express ICOSL but not ICOS (Figure 1a). For this purpose, we performed a linear scratch on a confluent monolayer of HaCat cells, which were then cultured in serum-free medium to minimize cell proliferation in the presence or absence of human ICOS-Fc or human ^F119S^ICOS-Fc (2 µg/mL), an ICOS-Fc mutant unable to bind ICOSL. After 24 h, microscopic evaluation revealed that treatment with ICOS-Fc but not ^F119S^ICOS-Fc led to a substantial increase in the percentage of migrating cells compared to that of the untreated control (Figure 1b,c). This result came as a surprise given that we had previously shown that ICOS-Fc inhibited migration of several cell types and wound closure in scratch assays performed on ECs and several tumor cell lines [8,9,10].

### 2.2. ICOS-Fc Treatment Accelerates Skin Wound Healing In Vivo

To assess the effect of ICOS-Fc on skin wound healing in vivo, skin wounds were created on the back of wild-type C57BL/6 mice, which were then daily instilled with 1x PBS with or without mouse ICOS-Fc. Wound healing was then followed up for 10 days. Consistent with our in vitro data, we found that treatment with ICOS-Fc significantly improved wound closure at days 1–6, while the healing curve gradually aligned with control levels at later time points (Figure 2a). Histological staining of fibroblasts and collagen performed at day 3 and 4 by H&E and picrosirius red staining, respectively, revealed that treatment with ICOS-Fc increased fibroblast migration into the wound compared to control at day 3 and, to a higher extent, day 4. In contrast, collagen deposition in ICOS-Fc-treated mice was similar to that of their control counterparts, indicating that treatment with ICOS-Fc favors repair but not scar formation (Figure 2b–d). Consistently, we observed a dramatic increase in αSMA gene expression, a marker of reparative myofibroblasts [19], at day 2 following treatment with ICOS-Fc, which decreased to control levels in the following days (Figure 2e).

Next, immunohistochemical staining of vessels with an anti-CD31 antibody revealed a significant increase in CD31^+^ vessels in mice treated with ICOS-Fc at day 3 and 4 compared to their control counterparts (Figure 3a), indicating enhanced wound angiogenesis. This was further supported by augmented CD31 and VEGF mRNA levels at day 1, both of which decreased to control levels in the subsequent days (Figure 3b).

To further characterize the healing process in wounded mice, we next sought to determine the infiltration extent of inflammatory cells by immunohistochemistry using antibodies specific for MPO, CD3, and F4/80. Results showed that treatment with ICOS-Fc decreased MPO^+^ neutrophils at day 3, whereas it increased CD3^+^ T cells at day 3 and 4, and F4/80^+^ monocyte/macrophages at day 3 (Figure 4a–c).

To better characterize the inflammatory microenvironment of the healing wound, we next assessed mRNA expression levels of IL-6, TNF-α, TGF-β, IL-33, IL-10, IL4, IFN-γ, OPN, TREM1, TREM2, ICOS, and ICOSL at day 1 to 3 by real time PCR. We found that treatment with ICOS-Fc strikingly increased expression of IL-6 at day 2, which decreased in the following days (Figure 5a). In contrast, expression of TNF-α was homogeneously decreased at all time points, while expression of TGF-β was moderately decreased at day 3 (Figure 5b,c). Expression of TREM1 and TREM2, respective markers of M1 and M2 macrophages, displayed opposite patterns since treatment with ICOS-Fc downregulated TREM1 and upregulated TREM2 at day 1, so that the TREM2/TREM1 ratio was increased about 5-fold (Figure 5d,e,g). ICOSL gene expression was decreased at day 3 upon ICOS-Fc treatment (Figure 5f), whereas no differences were detected for IL-10, IL-33, IL-4, IFN-γ, OPN, and ICOS (data not shown).

### 2.3. Effects of ICOSL Triggering on Macrophages

As our results demonstrated that treatment with ICOS-Fc increases the recruitment of macrophages to the healing wound, with apparent predominance of TREM2^+^ M2 macrophages, we sought to determine the effect of ICOS-Fc treatment on the migration of mouse M1 and M2 macrophages differentiated in vitro. To this end, spleen adherent cells were differentiated into macrophages by culturing them for 14 days in the presence of M-CSF. Cells were then cultured for an additional 2 days in the presence of IFN-γ to obtain M1 cells, or with IL-4 to obtain M2 cells; both culture conditions were performed in the presence or absence of LPS. At the end of the culture, differentiation was assessed by evaluating expression of NOS2 and ARG1 mRNA levels, marking M1 and M2 cells, respectively. As expected, M1 cells expressed higher levels of NOS2 and lower levels of ARG1 than M2 cells (Figure 6a). Analysis of ICOS and ICOSL mRNA showed that both M1 and M2 macrophages expressed ICOSL but not ICOS (data not shown), whereas M1 cells expressed higher ICOSL levels than M2 cells (Figure 6b).

These cells were then used to assess the effect of ICOS-Fc on cell migration induced by either CCL2 or OPN through Boyden chamber assay. To minimize the possible confounding effects due to interactions with Fcγ receptors (FcγRs), we used the recombinant ICOS-hFc, which consists of the extracellular portion of murine ICOS fused to the Fc of human IgG1. In addition, human ^F119S^ICOS-Fc, which does not bind to either ICOSL or mouse FcγRs, was used as negative control [8,9,10,11]. Consistent with our previous results in vivo, stimulation with ICOS-hFc increased the migration of M2 macrophages—regardless of the presence of LPS in the culture medium—compared to that of ^F119S^ICOS-Fc-treated cells. In contrast, ICOS-hFc treatment inhibited the migration of M1 macrophages stimulated with LPS, whereas it had no effect on those cultured in the absence of LPS. Similar results were observed by using either CCL2 (Figure 6c) or OPN as chemoattractant stimuli (Figure 6d). On the other hand, treatment with ^F119S^ICOS-Fc did not show any effect under any experimental conditions when compared to control migration assays performed in the absence of any form of ICOS-Fc.

### 2.4. Wound Healing in KO Mice

To determine the functional role of ICOS and ICOSL in wound healing in vivo, we investigated the effect of ICOS-Fc treatment in wounded mice deficient for ICOS or ICOSL, and NSG mice, lacking T, B, and NK cells.

Analysis of wound healing in the absence of ICOS-Fc treatment showed that ICOS^−/−^, ICOSL^−/−^, and NSG mice displayed a substantial healing delay, compared to wild type mice, starting from day 4 (Figure 7a). Treatment with ICOS-Fc significantly improved wound closure in ICOS^−/−^ and NSG mice, while it was ineffective in ICOSL^−/−^ mice (Figure 7b–d). Thus, the fact that ICOS-Fc treatment promotes wound healing in all strains expressing ICOSL but fails to do so in ICOSL^−/−^ mice suggests that ICOSL triggering drives tissue repair. Consistent with the data obtained in wild type mice, also in ICOS^−/−^ and NSG mice, ICOS-Fc significantly improved wound closure mainly in the initial part of healing (day 1–6), while the healing curve gradually aligned with control levels at later time points.

## 3. Discussion

The present study shows that ICOS and ICOSL cooperate in skin wound healing and that triggering of ICOSL by instillation of ICOS-Fc into the wound bed favors tissue repair in vivo. These results extend those obtained by Maeda et al. [15] showing that wound healing is delayed in ICOS^−/−^, ICOSL^−/−^, or ICOS/ICOSL^−/−^ mice, possibly due to defective production of IL-4, IL-10, and, especially, IL-6 at the wound site. Since this defective repair was overcome by adoptive transfer of wild-type T cells (expressing ICOS) in ICOS^−/−^ but not ICOSL^−/−^ mice, the authors concluded that the healing defect in KO mice could be ascribed to the impaired development of T helper type 2 cells due to the lack of ICOS-mediated co-stimulation of T cells.

Even though our findings confirm that wound healing is defective in mice lacking ICOS or ICOSL, the observation that ICOSL stimulation by ICOS-Fc is sufficient to accelerate the early phases of the healing process underscores the importance of ICOSL in ICOS/ICOSL-mediated tissue repair. Indeed, enhanced wound healing in response to ICOS-Fc treatment is readily apparent in both wild-type and ICOS^−/−^ mice, but not in mice lacking ICOSL, which indicates that this effect is not due to the inhibition of ICOS activity in T cells, but it is instead caused by ICOSL-mediated “reverse signaling” in other cell types. The fact that ICOS-Fc treatment is effective also in immunodeficient NSG mice confirms that T cells are not involved in ICOS-Fc-induced wound healing. Moreover, the lack of effect in ICOSL^−/−^ mice rules out possible confounding effects due to the potential interaction of ICOS-Fc with Fcγ receptors.

A key effect of ICOS-Fc is represented by increased angiogenesis and recruitment of fibroblasts at day 3 and 4, as judged by histologic analysis, both of which are preceded by upregulation of CD31 and VEGF-α—two markers of angiogenesis—and αSMA—a marker of reparative myofibroblasts—mRNA expression at day 1 and 2, respectively. Enhanced angiogenesis in response to ICOSL triggering was unexpected since previous works had shown that in vivo treatment with ICOS-Fc curbed neoplastic angiogenesis in several mouse tumor types, and in vitro experiments showed that ICOS-Fc had no effect on angiogenesis induced by VEGF whereas it inhibited that induced by OPN [8,18].

Another interesting observation from our histological analysis is that ICOS-Fc treatment can also modulate the infiltration of inflammatory cells by decreasing neutrophils and increasing T cells and macrophages. The decrease in neutrophils is in line with previous data showing that ICOS-Fc inhibits neutrophil adhesion to ECs, which may affect their recruitment into inflamed tissues [8]. The increase in T cells might be ascribable to the enhanced vascularization of the wound or to the functional antagonism between ICOS-Fc and ICOS expressed on T cells given that, at least in tumors, ICOS-Fc treatment is known to increase effector T cells and decrease regulatory T cells [18,20]. The increase in macrophages is quite intriguing as it is accompanied by a five-fold increase in TREM2/TREM1 expression ratio, which suggests that ICOS-Fc favors recruitment of TREM2^+^ M2-like reparative macrophages, as compared to TREM1^+^ M1-like inflammatory macrophages. This possibility is also supported by our cell migration experiments in vitro, showing that ICOS-Fc enhances the migration of M2 macrophages, whereas it inhibits that of M1 macrophages. The increased migration of M2 macrophages was unexpected, since ICOS-Fc had always inhibited the migration response of all cell types analyzed until then [8,9,10,12,20,21]. The different response of mouse M1 and M2 macrophages may be due to differences in their migration and adhesive properties likely caused by higher expression levels of β2 integrins in M1 vs. M2 cells [22]. Intriguingly, ICOS-Fc treatment also led to increased migration of keratinocytes, as judged by our scratch assay analysis, which could be the result of changes in size, shape, adhesiveness, and organization of keratin intermediate filaments of these cells as shown previously [23].

Overall, the effects of ICOS-Fc on wound healing are in line with our previous work showing that CCl_4_-treated ICOS^−/−^ or ICOSL^−/−^ mice develop a more severely acute inflammatory liver damage, along with a reduction of reparative macrophages, compared to their wild-type counterparts. Moreover, treatment with ICOS-Fc protected ICOS-deficient mice from this increased damage, simultaneously restoring the number of reparative macrophages, whereas it had no effects in ICOSL^−/−^ mice [17]. These findings are also in line with the aforementioned study by Maeda et al. [15], showing that mice lacking ICOS and/or ICOSL display decreased angiogenesis and a reduction of T cells and macrophages at the wound site. Intriguingly, the authors observed decreased IL-6 in the wounds of these mice, and local application of exogenous IL-6 in the initial phase of healing (day 1) led to a substantial improvement of tissue repair. A potential role of ICOSL-induced IL-6 production in wound healing is also supported by our observation that treatment with ICOS-Fc of wounded wild-type mice increases the expression of IL-6 at day 2 [15].

## 4. Materials and Methods

### 4.1. Scratch Assay

HaCat cells (human keratinocytes) were purchased from ATCC (Manassas, VN, USA) and grown in DMEM (Life Technologies, Carlsbad, CA, USA) medium plus 10% fetal bovine serum (FBS; Life Technologies). HaCat cells were plated in six-well plates at a concentration of 10^6^ cells/well and grown to confluence. To prevent cell proliferation, cells were incubated for 12 h in FBS-free medium. Cell monolayers were wounded by scratching with a sterile plastic pipette tip along the diameter of the well. Cells were then incubated in culture medium in the absence or presence of 2 µg/mL human ICOS-Fc or ^F119S^ICOS-Fc, an ICOS-Fc mutant unable to bind ICOSL. To monitor cell migration in the wound, five fields of each wound were analyzed and photographed immediately after scratching (0 h) and 24 h later. The wound closure was calculated with the following formula: (1 − (scratch width of the treated group/scratch width of the control group)) × 100%.

ICOS and ICOSL expression was assessed by immunofluorescence and flow cytometry (Attune NxT, Thermo-Fisher, Waltham, MA, USA) using PE-conjugated mAb to ICOS or ICOSL (R&D System, Minneapolis, MN, USA). The mean fluorescence intensity ratio (MFI-R) was calculated according to the following formula: MFI of the stained sample histogram (arbitrary units)/MFI of the control histogram (arbitrary units).

### 4.2. Mice

C57BL6/J (WT), NOD-SCID-IL2R γ-null mice (NSG) and knockout B6.129P2-Icos^tm1Mak/^J (ICOS^−/−^) and B6.129P2-Icosl^tm1Mak^/J (ICOSL^−/−^) mice (The Jackson Laboratory, Bar Harbor, ME, USA) were bred under pathogen-free conditions in the animal facility at Università del Piemonte Orientale, Department of Health Sciences (Authorization No. 217/2020-PR) and treated in accordance with the Ethical Committee and European guidelines.

### 4.3. In Vivo Wounds

The day before wound induction (day-1), WT, NSG, ICOS^−/−^, and ICOSL^−/−^ mice were anesthetized with 2% isoflurane and their back was shaved. At day 0, mice were anesthetized as above, and wounds were made on their back using a 4 mm puncher (Kai Medical, Solingen, Germany). The wound area was photographed and measured using the following formula: (a/2) × (b/2) × 3.14, where “a” and “b” are the two perpendicular diameters. In the following days, wound closure was calculated using the following formula: (wound area^T0^-wound area^TX^)/wound area^T0^ × 100. Mice were treated daily with 10 µg/mouse ICOS-Fc in PBS instilled directly into the wound site; controls were treated with an equal volume of PBS. Mice were monitored daily for 12 days, at which point in time the wound was closed. In some experiments, mice were sacrificed at day 1, 2, 3, and 4 to harvest and analyze the healing tissue. Each experiment involved 4–7 mice for each condition tested; each condition was tested in 2–3 independent experiments. Sample size was calculated using G*Power (RRID:SCR_013726) software (Power: 80%; Significance: 95%).

### 4.4. Real-Time PCR Analysis

Total RNA was isolated from skin samples collected at day 1, 2, and 3 post-injury, or from in vitro-differentiated macrophages using TRIzol reagent (Sigma-Aldrich, St. Louis, MO, USA). RNA (1 µg) was retro-transcribed using QuantiTect Reverse Transcription Kit (Qiagen, Hilden, Germany). Expression of the IL-6, TNF-α, TGF-β, IL-33, IL-10, IL-4, IFN-γ, OPN, TREM1, TREM2, VEGF-α, α-SMA, ICOS, NOS2, ARG1, and ICOSL mRNA were evaluated by real-time PCR (Assay-on Demand; Applied Biosystems, Foster City, CA, USA). The β-actin gene was used to normalize the cDNA amounts. Real-time PCR was performed using the CFX96 System (Bio-Rad Laboratories, Hercules, CA, USA) in duplicate for each sample in a 10 µL final volume containing 1 µL of diluted cDNA, 5 µL of TaqMan Universal PCR Master Mix (Applied Biosystems, Foster City, CA, USA), and 0.5 µL of Assay-on-Demand mix. The results were analyzed with a ΔΔ threshold cycle method.

### 4.5. Histological Analysis

Skin samples were collected at day 3 and 4 post-injury and processed for paraffin embedding. Samples were cut at 4-µm thickness and stained with hematoxylin and eosin (H&E) (Sigma-Aldrich) for tissue morphology and fibroblast evaluation, or with picrosirius red (Abcam, Cambridge, UK) to evaluate the extent of fibrosis.

Immunohistochemical staining of CD31, MPO, CD3, and F4/80 was performed to detect neo vessel formation and infiltration of immune cells (i.e., neutrophils, T cells, and macrophages). Samples were treated with citrate buffer (Vector Laboratories, Burlingame, CA, USA) for antigen retrieval, and endogenous peroxidases were blocked with 3% H_2_O_2_ (Sigma-Aldrich). To avoid secondary antibody unspecific binding, samples were pre-incubated with 5% normal goat serum (NGS) (Sigma-Aldrich) for 1 h at room temperature (RT). Samples were stained with rabbit antibodies against CD31 (Abcam, 1:50), MPO (Invitrogen, 1:100), CD3 (Invitrogen, 1:150), or F4/80 (Invitrogen, 1:100) overnight at 4 °C and, then, with a goat anti-rabbit horseradish peroxidase (HRP)-conjugated secondary antibody (Sigma-Aldrich), followed by 3,3′-diaminobenzidine (DAB) (Agilent Dako, Santa Clara, CA, USA). Successively, samples were counterstained with hematoxylin (Sigma-Aldrich), dehydrated, and mounted on cover slips. Slides were acquired using Pannoramic MIDI (3D Histech, Budapest, Hungary) at 200× magnification. The positive areas for CD31, fibroblasts, and collagen were calculated using the following formula: (positive area/total area) × 100%. MPO, CD3, and F4/80 positive cells were expressed as cell number/field counted in 15 fields for each sample.

### 4.6. Macrophage Migration Assay

Spleen cells were separated by density gradient centrifugation using the Ficoll-Hypaque reagent (Lympholyte-M, Cedarlane Laboratories, Burlington, ON, Canada) and incubated in tissue culture dishes for 2 h with DMEM supplemented with 10% FBS. Subsequently, supernatants and non-adherent cells were discarded, and the adherent cells were rinsed three times and cultured in DMEM (Life Technologies) medium supplemented with 10% FBS, 1% glutamine, and 1% penicillin/streptomycin plus 20 ng/mL M-CSF (Immunotools, Friesoythe, Germany) for 14 days (normal DMEM medium). At day 14, adherent cells were cultured for additional 48 h with interferon-γ (IFN-γ; 100 U/mL Immunotools) to obtain M1 macrophages, and with interleukin-4 (IL-4; 20 ng/mL Immunotools) to obtain M2 macrophages; each culture condition was performed in the presence or absence of LPS (LPS; 100 ng/mL Sigma).

Macrophage migration was assessed by the Boyden chamber migration assay (BD Biosciences, San Jose, CA, USA). Cells were plated (10^4^ cell/well) onto the apical side of 50 µg/mL Matrigel-coated filters in serum-free medium in the presence or absence of msICOS-huFc (2 µg/mL), composed by the extracellular portion of murine ICOS fused to the Fc of human IgG1, or human ^F119S^ICOS-Fc (2 µg/mL). Mouse CCL2 (30 nM, Immunotools) or OPN (10 µg/mL) were used as chemoattractants in the bottom chamber. After 6 h, the cells on the apical side were wiped off with Q-tips. Cells on the bottom of the filter were stained with crystal violet and all counted (quadruplicate filter) with an inverted microscope. Data are shown as number of migrating cells [12].

### 4.7. Statistical Analyses

Statistical analyses were performed using Mann–Whitney U test, Wilcoxon test, Dunnett’s test, or Student’s *t*-test using GraphPad Instat Software (GraphPad Software, San Diego, CA, USA), as indicated. Data are expressed as mean and standard error of the mean (SEM) and statistical significance was set at *p* < 0.05.

## 5. Conclusions

In conclusion, this work shows that ICOSL plays a key role in wound healing and that triggering of ICOSL by ICOS-Fc favors healing by increasing angiogenesis and the recruitment of fibroblasts and reparative macrophages. Therefore, ICOS-Fc and other molecules capable of triggering ICOSL might be exploited to improve wound closure in patients with impaired tissue repair.

## Figures and Tables

**Figure 1 ijms-23-07363-f001:**
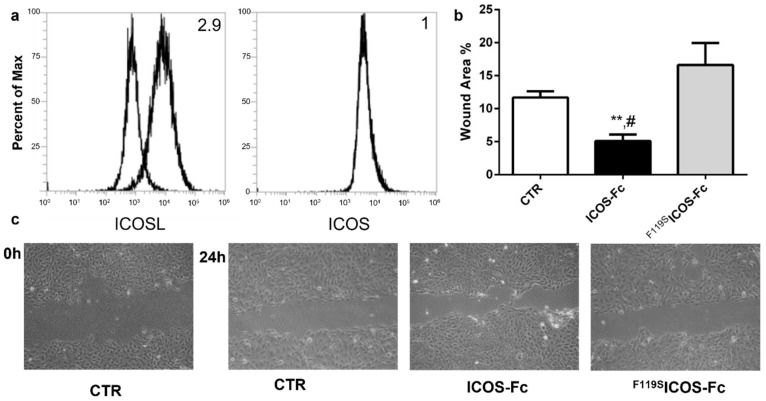
Effect of ICOS-Fc stimulation on the motility of HaCat cells by scratch assay. HaCat cells were cultured to confluence on 6-well plates. A scratch was made through the cell layer using a pipette tip and cells were then cultured in the presence or absence of 2 µg/mL ICOS-Fc or ^F119S^ICOS-Fc for 24 h. (**a**) ICOS and ICOSL expression in HaCat cells. (**b**) Wound area % after 24 h of treatment as above, calculated as: 1 − (scratch width of the treated group/scratch width of the control group) × 100; results are the means from three independent experiments; ** *p* < 0.01 vs. CTR; # *p* < 0.05 vs. ^F119S^ICOS-Fc, calculated by paired *t*-test. (**c**) Representative microphotographs of the wounded area taken immediately after the scratch was made 0 h and 24 h later to monitor cell migration into the wounded area (original magnification 10×).

**Figure 2 ijms-23-07363-f002:**
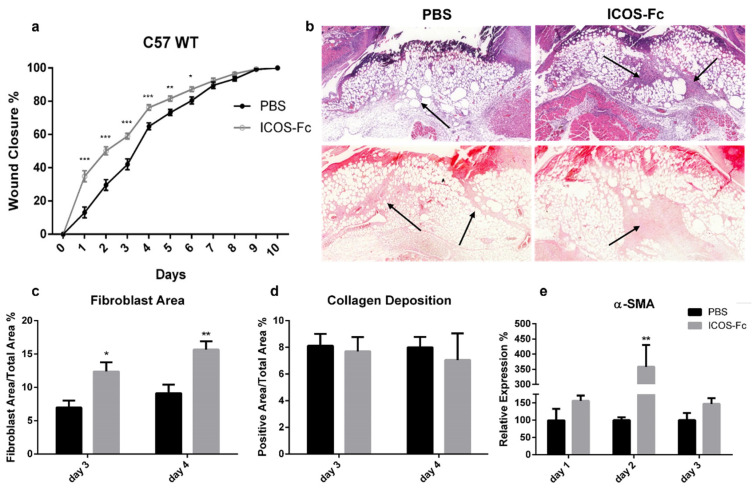
Effect of treatment with ICOS-Fc on wound healing in wild-type C57BL/6 mice. (**a**) Wound healing % in mice treated with PBS (n = 21) or ICOS-Fc (n = 22) calculated as (wound area^T0^-wound area^TX^)/wound area^T0^ × 100%; mean ± SE; (**b**) representative microphotographs of the staining (magnification 200×) at day 4; upper panels: Hematoxylin/eosin (H&E) for fibroblast area quantification (black arrows); lower panels: picrosirius red for collagen deposition quantification (black arrows). (**c**,**d**) Wound area % occupied by fibroblasts and collagen as detected by H&E and picrosirius red staining, respectively, at day 3 and day 4. (**e**) αSMA mRNA expression analysis by real time PCR at day 1, day 2, and day 3. Results are expressed as mean ± SE from 4 independent experiments; * *p* < 0.05; ** *p* < 0.005; *** *p* < 0.001, calculated by Mann–Whitney test.

**Figure 3 ijms-23-07363-f003:**
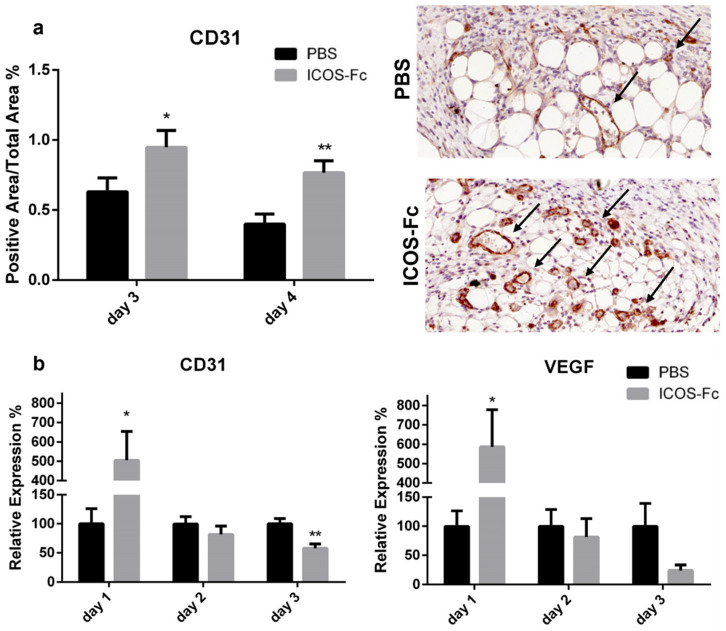
Treatment with ICOS-Fc stimulates angiogenesis during wound healing. (**a**) Wound area occupied by CD31^+^ vessels as detected by immunohistochemistry. Left panel: results at day 3 and day 4 expressed as mean ± SE from four independent experiments. Right panels: Representative microphotographs of CD31 staining (magnification 200×) at day 4. (**b**) Expression of the CD31 (left) and VEGF (right) mRNA levels as assessed by real time PCR at day 3 and day 4 and expressed as mean ± SE from four independent experiments. Results are expressed as % of the mRNA amount detected in PBS-treated mice at each time point; * *p* < 0.05; ** *p* < 0.005; calculated by unpaired Student’s *t*-test.

**Figure 4 ijms-23-07363-f004:**
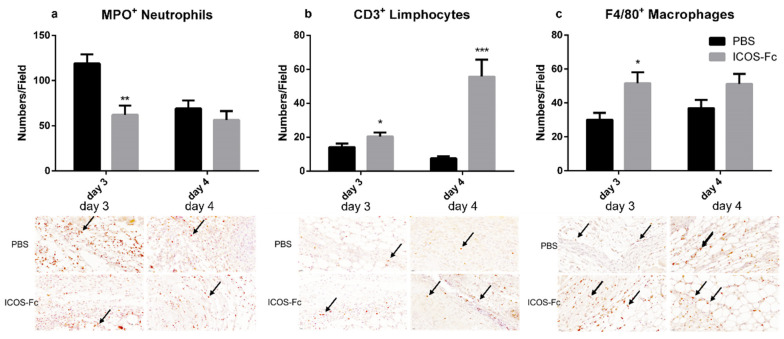
Effect of treatment with ICOS-Fc on the infiltration of inflammatory cells in the wound bed. Infiltration by neutrophils, T cells, and macrophages was assessed by immunohistochemistry using antibodies against MPO (**a**), CD3 (**b**), and F4/80 (**c**), respectively. Upper panels: number of positive cells per field counting 9 fields in each experiment at day 3 and day 4; results are expressed as mean ± SE. Lower panels: representative immunohistochemical staining (magnification 400×) at day 3. Statistical analysis was performed with Mann–Whitney test: * *p* < 0.05; ** *p* < 0.01; *** *p* < 0.0005.

**Figure 5 ijms-23-07363-f005:**
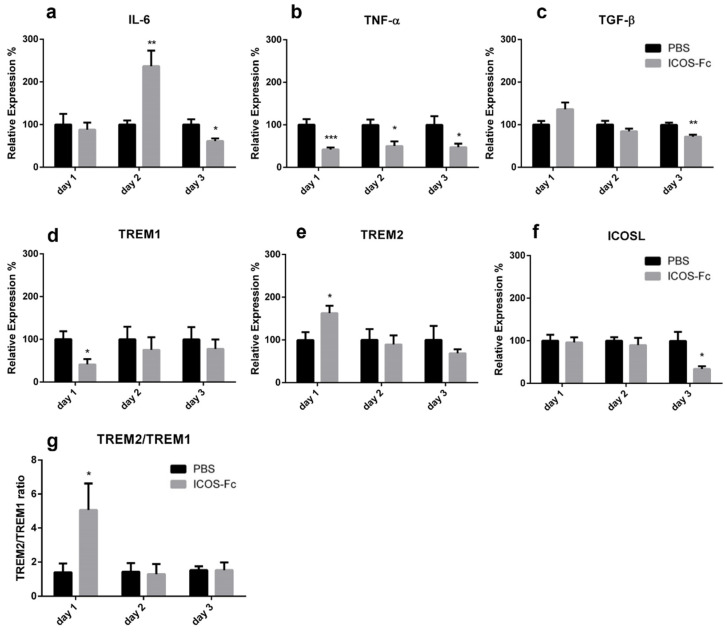
Effect of treatment with ICOS-Fc on the expression of inflammatory molecules in the wound. Expression of IL-6 (**a**), TNF-α (**b**), TGF-β (**c**), TREM1 (**d**), TREM2 (**e**), and ICOSL (**f**) mRNA as assessed by real time PCR at day 1, day 2, and day 3, and expressed as mean ± SE from 3 independent experiments. (**g**) TREM2/TREM1 ratio. Results are expressed as % of the mRNA amount detected in PBS-treated mice at each time point; * *p* < 0.05; ** *p* < 0.005; *** *p* < 0.001, calculated by unpaired Student’s *t*-test.

**Figure 6 ijms-23-07363-f006:**
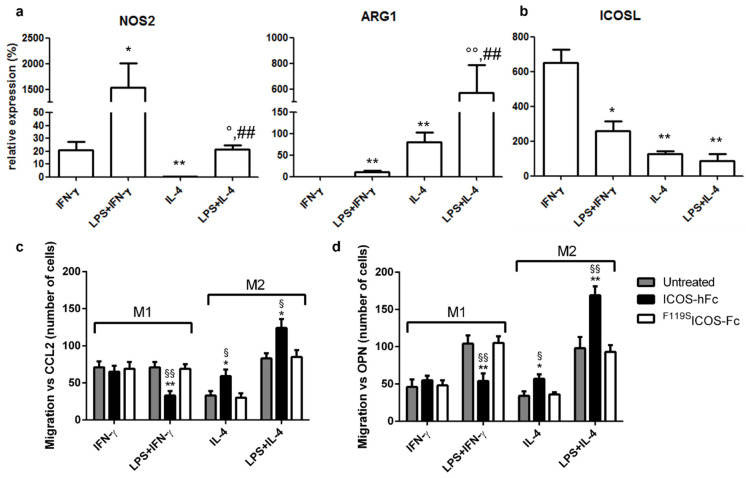
Effect of ICOS-Fc on the migration of murine M1 and M2 macrophages. Macrophages obtained by culturing adherent spleen cells with M-CSF for 14 days (M0) were polarized to M1 or M2 macrophages by culturing them with IFN-γ or LPS + IFN-γ (M1) or with IL-4 or LPS + IL-4 (M2) for 48 h. (**a**) NOS2, ARG1, and (**b**) ICOSL gene expression analysis by real-time PCR. Values are expressed as % of the mRNA detected in M0 macrophages stimulated with LPS for 48 h (* *p* <0.05, ** *p* < 0.01 vs. IFN-γ, ° *p* < 0.05, °° *p* < 0.01 vs. IL-4, ## *p* < 0.01 vs. LPS + IFN-γ, Mann–Whitney test). (**c**,**d**) Cell migration assay. Murine M1 and M2 macrophages were cultured in the presence and absence of ICOS-hFc or ^F119S^ICOS-Fc using either CCL2 (30 nM) (**c**) or OPN (10 µg/mL) (**d**) as chemotactic factors. Values are expressed as number of migrating cells, stimulated with either CCL2 or OPN. The results are expressed as mean ± SE from n = 3–8 experiments; differences versus either ^F119S^ICOS-Fc (* *p* < 0.05; ** *p* < 0.01) or the untreated control (^§^ *p* < 0.05; ^§§^ *p* < 0.01) for each condition are calculated by Dunnett test.

**Figure 7 ijms-23-07363-f007:**
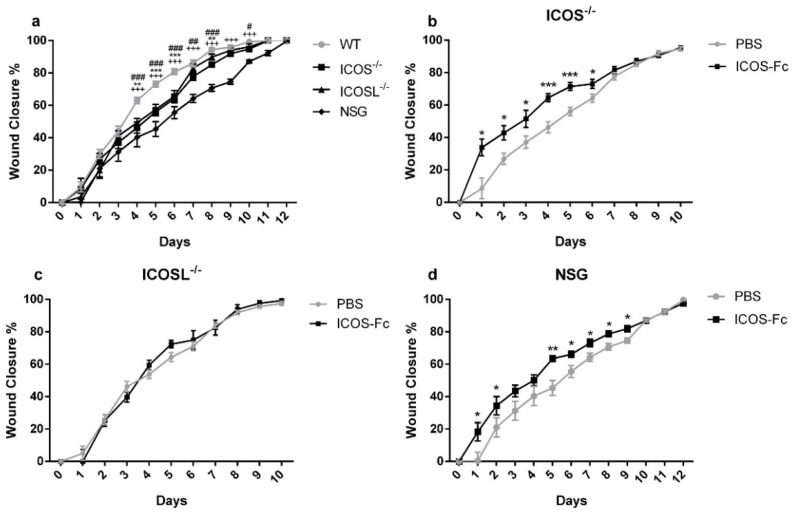
Effect of treatment with ICOS-Fc on wound healing in various KO mice. Wound healing % was calculated as described in the legend to Figure 2. (**a**) Comparison of wound healing in wild-type mice (n = 21), ICOS^−/−^ (n = 16), ICOSL^−/−^ (n = 18), and NSG (n = 15) mice. (**b**–**d**) Effect of treatment with ICOS-Fc on wound healing in ICOS^−^^/−^ (PBS: n = 16; ICOS-Fc: n = 11), ICOSL^−^^/−^ (PBS: n = 9; ICOS-Fc: n = 9), and NSG (PBS: n = 15; ICOS-Fc: n = 16) mice. * *p* < 0.05; ** *p* < 0.005; *** *p* < 0.001; # *p* < 0.05; ## *p* < 0.005; ### *p* < 0.001; +++ *p* < 0.001; calculated by Mann–Whitney test.

## Data Availability

Raw data are available on request from the corresponding author.

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
