# Peer review of "ICOSL Stimulation by ICOS-Fc Accelerates Cutaneous Wound Healing In Vivo"

_ijms, 2022, doi:10.3390/ijms23137363_

Round 1

Reviewer 1 Report

In the manuscript, “ICOSL stimulation by ICOS-Fc accelerates cutaneous wound healing

in vivo” by Stoppa et al., the authors investigate the effect of the ectopic treatment of cutaneous wounds with an ICOS-Fc construct in mice.

The author show that such a construct enhances the motility of HaCaT cells in vitro in a scratch assay and enhances wound healing in mice. The ICOS-Fc molecule enhanced wound closure in wt and in ICOS deficient mice, but not in ICOS-L deficient control mice. Most interestingly, treatment with the ICOS-Fc molecule had also effects on wound closure in severely immune deficient NSG mice; strongly suggesting that the effect is not T-cell mediated but mediated by ICOS-L expression on parenchymal cells.

The experiments appear to be performed in a technical sound way and are novel.

There are just two minor aspects, I would like the authors to address:

-       I don’t really understand, why the Figure labels, show “T” for days. It might be easier, if the authors directly wrote “day 1”, “day 2” and so on.

-       In Figure 6c and 6d, we are only shown highly processed data, which could be deceiving. Therefore, I would strongly suggest that raw data of migration are shown as well and only then the ratio data.

Author Response

The author show that such a construct enhances the motility of HaCaT cells in vitro in a scratch assay and enhances wound healing in mice. The ICOS-Fc molecule enhanced wound closure in wt and in ICOS deficient mice, but not in ICOS-L deficient control mice. Most interestingly, treatment with the ICOS-Fc molecule had also effects on wound closure in severely immune deficient NSG mice; strongly suggesting that the effect is not T-cell mediated but mediated by ICOS-L expression on parenchymal cells.

The experiments appear to be performed in a technical sound way and are novel.

There are just two minor aspects, I would like the authors to address:

-       I don’t really understand, why the Figure labels, show “T” for days. It might be easier, if the authors directly wrote “day 1”, “day 2” and so on.

Reply: As suggested by the reviewer, “T” have been substituted with “day” in the text and figures

-       In Figure 6c and 6d, we are only shown highly processed data, which could be deceiving. Therefore, I would strongly suggest that raw data of migration are shown as well and only then the ratio data

Reply: As suggested by the reviewer, Panel 6C has been modified in order to show absolute numbers of migrating cells

Reviewer 2 Report

The authors described "ICOSL Stimulation by ICOS-Fc Accelerates Cutaneous Wound Healing In Vivo" using animal study. This topic may be attractive for potential readers. The study design is good, however, I have some concerns and suggestions to improve this manuscript.

1. Introduction part is too long, especially we do not need former part by line 60.

2.  After all, treatment with ICOS-Fc protected ICOS-deficient mice from this increased damage, simultaneously restoring the number of reparative macrophages, whereas it had no effects in ICOSL-/- mice.  The authors described "Our results show that ICOS-Fc improves would healing likely by increasing angiogenesis" in Introduction. However, they did not mention the results of angiogenesis. Which is correct?

3. English editing should be needed.

4. I understood that treatment with ICOS-Fc significantly improved wound closure in ICOS-/- and NSG mice, while it was ineffective in ICOSL-/- mice. However, the healing time was similar and not different around 10 days. Please mention the time of wound healing.

5. How did you decide the sample size of mice?

Author Response

Comments and Suggestions for Authors

The authors described "ICOSL Stimulation by ICOS-Fc Accelerates Cutaneous Wound Healing In Vivo" using animal study. This topic may be attractive for potential readers. The study design is good, however, I have some concerns and suggestions to improve this manuscript.

  1. Introduction part is too long, especially we do not need former part by line 60.

Reply: The initial part of the Introduction has been omitted, as suggested by the reviewer.

  1. After all, treatment with ICOS-Fc protected ICOS-deficient mice from this increased damage, simultaneously restoring the number of reparative macrophages, whereas it had no effects in ICOSL-/- mice.  The authors described "Our results show that ICOS-Fc improves would healing likely by increasing angiogenesis" in Introduction. However, they did not mention the results of angiogenesis. Which is correct?

Replay: Data suggesting that ICOS-Fc supports angiogenesis are reported in Fig.3, showing that treatment with ICOS-Fc increases blood vessel formation and expression of CD31 and VEGF in vivo. We did not perform angiogenesis experiments in vitro since our previous work showed that that ICOS-Fc had no effect on in vitro angiogenesis induced by VEGF and it inhibited that induced by OPN. To make clear the point, the final sentence of the Introduction has been changed with "Our in vivo results show that ICOS-Fc improves would healing likely by increasing angiogenesis", and the third paragraph of the Discussion section discussing this point has been extended as it follows: Enhanced angiogenesis in response to ICOSL triggering was unexpected since previous works had shown that in vivo treatment with ICOS-Fc curbed neoplastic angiogenesis in several mouse tumor types, and in vitro experiments showed that ICOS-Fc had no effect on angiogenesis induced by VEGF whereas it inhibited that induced by OPN [8,18]”      

  1. English editing should be needed.

Replay: English has been revised by a professional English translator (http://www.abeschool.it/)

  1. I understood that treatment with ICOS-Fc significantly improved wound closure in ICOS-/- and NSG mice, while it was ineffective in ICOSL-/- mice. However, the healing time was similar and not different around 10 days. Please mention the time of wound healing.

Replay: As suggested by the reviewer, we mentioned the point in the description of Fig. 7 data in the Results section by adding the following sentence: “Consistent with the data obtained in wild type mice, also in ICOS-/- and NSG mice, ICOS-Fc significantly improved wound closure mainly in the initial part of healing (day 1-6), while the healing curve gradually aligned with control levels at later time points”.

  1. How did you decide the sample size of mice?

Each experiment involved 4-7 mice for each condition tested, in order to allow intraexperimental comparisons; each condition was tested in 2-3 independent experiments, in order to test interexperimental reproducibility. In the revised manuscript, this point has been stated in the Material and Methods section.

Round 2

Reviewer 2 Report

The authors didn’t mention the sample size. I mean how they decide the sample size. Using calculator or reference? As this study was an animal study, this point is critical.

Author Response

The authors didn’t mention the sample size. I mean how they decide the sample size. Using calculator or reference? As this study was an animal study, this point is critical.

Reply: Sample size was calculated using G*Power (RRID:SCR_013726) software (Power: 80%; Significance: 95%). This point has been reported in the Materials and Methods section of the revised manuscript.